# Physicochemical and In Vitro Starch Residual Digestion Structures of Extruded Maize and Sorghum Starches Added with Sodium Stearoyl Lactylate

**DOI:** 10.3390/foods12101988

**Published:** 2023-05-14

**Authors:** Julian de la Rosa-Millan, Erick Heredia-Olea, Esther Pérez-Carrillo, Raquel Peña-Gómez, Sergio O. Serna-Saldívar

**Affiliations:** 1Bio-Foods Research Lab., Escuela de Ingenieria y Ciencias, Tecnologico de Monterrey, Av. Eugenio Garza Sada 2501 Sur, Monterrey C.P. 64849, Mexico; 2Centro de Biotecnologia FEMSA, Escuela de Ingenieria y Ciencias, Tecnologico de Monterrey, Av. Eugenio Garza Sada 2501 Sur, Monterrey C.P. 64849, Mexico; erickho@tec.mx (E.H.-O.); perez.carrillo@tec.mx (E.P.-C.); a00819923@tec.mx (R.P.-G.); sserna@tec.mx (S.O.S.-S.)

**Keywords:** extrusion, starch, corn, sorghum, in vitro digestion, molecular structure, SSL

## Abstract

This research aimed to characterize the physicochemical, in vitro digestion, and structural features of digestion residues of maize and sorghum starches subjected to thermoplastic extrusion, along with the influence of Sodium Stearoyl Lactylate (SSL), to obtain improved starches for food applications and to understand their behavior when consumed as a food ingredient. The morphology of the extruded materials showed remanent starch granules when SSL was used. A higher amount of medium and large linear glucan chains were found in these particles, influencing higher thermal stability (ΔH ≈ 4 J/g) and a residual crystallinity arrangement varying from 7 to 17% in the extrudates. Such structural features were correlated with their digestibility, where slowly digestible starch (SDS) and resistant starch (RS) fractions ranged widely (from 18.28 to 27.88% and from 0.13 to 21.41%, respectively). By analyzing the data with a Principal component analysis (PCA), we found strong influences of B2 and B3 type chains on the thermal stability of the extrudates. The amylose and smaller glucan chains (A and B1) also significantly affected the emulsifying and foam stability properties. This research contributes to the molecular knowledge of starch in extruded products with broad food applications.

## 1. Introduction

Starch is one of the most abundant polysaccharides in nature and is crucial for human nutrition. Practically all over the globe, starch-rich foods and ingredients constitute the main staples for humankind, cereals being one of the most used sources for its isolation [1]. Industrial starch production has a bright future in sustainable food supply to the world; however, it presents some structural stability limitations under extreme pH (acid or basic) and high shear conditions [2]. Under processing, starch pasting and retrogradation after cooking (and/or storage) are the principal limitations restricting its use in food products [1]. Because of this, chemical, physical and enzymatic methods have been employed to improve and broaden their use in diverse food systems, where green, safe and efficient modifications would be favored by producers and consumers [2]. Maize starch is an ingredient widely used in food product development, mainly to provide structure to foodstuffs, it is relatively abundant and has a low cost compared with other sources [1]. For this, native maize starch is purposely modified to improve physicochemical and functional properties, diversifying its use. Some commonly used physical techniques include annealing, gelatinization, and thermal moisture treatments, with the disadvantage of decreasing their water absorption properties and solubility [3]. On the other hand, sorghum starch has similar physicochemical characteristics to maize, with the advantage that this seed has emerged as an alternative because it is a drought-tolerant crop that can grow in high salinity soil, being beneficial to developing countries. Thermo-mechanical approaches, such as extrusion, have been evaluated to avoid chemical reagents that may restrict their usage; Panyoo and Emmambux [4] observed that the extrusion cooking of maize starch with a standard screw and stearic acid (1.5% or 4%) significantly (*p* < 0.05) decreased the WAI and first peak viscosity and gel texture of extrudates. Moreover, the extrusion cooking of maize starch at a screw speed of 125 rpm yielded more amylose–lipid complexes. Other investigations have tested emulsifiers aimed to retard staling on intermediate-moisture bread due to molecular interactions with starch, enabling single helical V-type complexes with amylose, reducing starch swelling and amylose leaching, avoiding migration of water from gluten, and diminishing starch crystallization [3]. This complex-forming ability relies on amylose structure, the type, pH, and ion concentration of emulsifiers. However, this may not be the only reason for the anti-stalling effect of emulsifiers [5]. Extrusion is a process that includes various operations, such as mixing, conveying, heating, kneading, shearing, and shaping. During this continuous process, the thermal energy generated combined with the shearing effect may induce physicochemical changes and molecular transitions, such as starch gelatinization and dextrination, which could be beneficiated from chemical interactions promoted by stabilizers [6]. Complimentary food additives have been used to improve the quality of extruded products. Wang, An, Jin, Xie, Zhuang, and Kim [7] evaluated the effect of sodium stearoyl lactylate (SSL) on extrudate instant rice and observed a positive impact on cooking properties. Interestingly, they attributed this change to protein-SSL-starch interactions. Even though the SSL and extrusion are affected, maize and sorghum starches are used as brewing adjuncts for lager beer production [8,9], and the techno-functionality and starch molecular changes due to extrusion have not been further studied. This work aims to analyze the effects of thermoplastic extrusion in the presence of SSL on maize and sorghum starches, especially in terms of physicochemical properties and at the molecular level in their digestion residues.

## 2. Materials and Methods

### 2.1. Materials

Maize and sorghum starch were provided by Cervecería Cuauhtemoc Moctezuma, Heineken’s subsidiary in Mexico. Sodium Stearoyl Lactylate (SSL) was provided by Paalsgard (San Luis Potosí, México). All other reagents were laboratory grade.

All analyses were performed in triplicate.

### 2.2. Starch Extrusion

Native corn starch was extruded using a twin-screw corotating extruder (BCTM-30, Bühler, Switzerland) with a 600 mm length and L/D = 20 at a volumetric feed rate of 4 kg/h (calibrated with sugar; 8.50 Kg of starch/h with a wet basis moisture content of 11.30%). The die diameter was 5 mm, and the screw configuration was selected specifically to create high levels of shear. The first section contained only conveying elements, with the next containing conveying and kneading elements. Finally, the high-shear section contained conveying, reverse conveying, and kneading elements. The temperature was controlled at the final stage of the extruding chamber by using a TT-137 N water heater (Tool-temp, Sulgen, Switzerland). The variables where feedstock tempering moisture (min = 20%, max = 40%), temperature (min = 60 °C, max = 100 °C), screw speed (min = 100 rpm, max = 180 rpm), and SSL content (min = 0%, max = 1%) (Table 1).

In the case of sorghum starch, we consider reducing the experimental runs due to our preliminary data, which showed close similarities with maize; in this case, the screw speed was fixed at 140 RPM, and the variables tempering moisture: 20% or 40%; temperature: 80 °C or 100 °C; %SSL: 0.5% or 1% were used (Table 2). Before extrusion, starch/SSL mixtures were stored for 24 h (2 kg per treatment). All treatment extrudates were dried in a convection oven at 50 °C until the moisture level reached ≈13%. The dried extrudates were milled (Thomas Wiley Mill, Thomas Scientific, Swedesboro, NJ, USA) through a 2 mm screen and immediately stored at room temperature until tested.

### 2.3. Starch Granule Morphology

Starch granules’ morphology and birefringence patterns were observed under normal and polarized light with an Olympus CX-33 microscope (Hong Kong, China) equipped with polarized filters. The images were recorded at 40× magnification.

### 2.4. Starch and Amylose Contents

Total starch (K-TSTA) and amylose (K-AMYL) contents were determined using Megazyme kits (Megazyme, Wicklow, Ireland) using 500 mg of sample, according to the method reported by de la Rosa-Millan et al. [10]. The total starch procedure is based on the amylolysis of starch granules using a thermostable α-amylase, rendering soluble branched and unbranched maltodextrins, later a glucoamylase converts those molecules to glucose that is quantified via glucose oxidase/peroxidase enzyme (GODPOD). In the case of amylose quantification, the starch samples were dissolved and reacted with Concanavalin-A (Con-A), which selectively bounds to the reducing end of amylopectin molecules, creating a complex that facilitates its precipitation. Later, the supernatant was recovered and hydrolyzed with a mixture of α-amylase and glucoamylase, rendering glucose molecules quantified with GODPOD.

### 2.5. Thermal Properties

All samples were analyzed with a differential scanning calorimeter (Diamond DSC, Perkin Elmer, Norfolk, VA, USA), using the conditions and procedure described by de La Rosa-Millán et al. [10]. The onset temperature (To), peak temperature (Tp), conclusion temperature (Tf), and gelatinization enthalpy (ΔH) were calculated with the Pyris software (Perkin Elmer, Norwalk, CT, USA). The gelatinization degree was calculated as shown in Equation (1) where ΔH1 was the gelatinization enthalpy of the native starch sample, and ΔH2 was the gelatinization enthalpy of the corresponding treatment.
Gelatinization degree (%) = (ΔH1 − ΔH2)/ΔH1 × 100(1)

### 2.6. X-ray Diffraction

The diffraction patterns of starches were obtained using an Advance D8 diffractometer (Bruker, Coventry, UK) at 35 kV with a CuK-α radiation source (1.542 Å). The samples were scanned in the angular range of 4–35° (2Ɵ). The crystallinity percentage (%C) was determined from the diffractograms calculating the area corresponding to the crystalline peaks (Ap); from the difference between the area under the curve and the area of the amorphous halo, the total area under the curve (At), and the instrumental noise (N) according to Equation (2)
%C = Ap/(At − N)(2)

### 2.7. Starch Digestion Fractions

All samples’ in vitro digestion fractions were determined according to the Englyst, Kingman and Cummings [11] protocol. For this, 400 mg (d.b.) of each material was hydrated with 10 mL of de-ionized water and heated in a boiling water bath for 20 min with vortexing every 5 min. The tubes were cooled at 37 °C, and 8 mL of pepsin dispersion (5.21 mg/mL) was added and incubated in a shaking water bath at 37 °C. Then, 8 mL of 0.5 M sodium acetate buffer (pH 5.2) was added and homogenized, and then 4 mL of an enzyme solution (pancreatin, amyloglucosidase, and invertase) and seven glass beads (7 mm diameter). After 20 and 120 min of reaction, 1 mL aliquots were taken and mixed with 2 mL of 80% ethanol; the glucose content was quantified with the glucose oxidase-peroxidase reagent. Starch classifications based on the rate of hydrolysis were calculated with the following equations: rapidly digestible starch (RDS) (digested within 20 min) (Equation (3)), slowly digestible starch (SDS) (Equation (4)), and resistant starch (RS) (Equation (5)). Where G20 and G120 are weights of reducing sugars after digestion at 20 min and 120 min, respectively; FG is the weight of reducing sugar before, whereas TS is the weight of total starch in the sample.
RDS (%) = (G20 − FG) × 0.9 × 100/TS(3)
SDS (%) = (G120 − G20) × 0.9 × 100/TS(4)
RS (%) = 100% − SDS − RDS(5)

### 2.8. In Vitro Starch Digestion Rate and Predicted Glycemic Index

The method described by Goñi, García-Alonso and Saura-Calixto [12] was employed to evaluate the in vitro rate of starch hydrolysis in cooked dispersions. The percentage of hydrolyzed starch by Porcine pancreatic α-amylase at 30, 60, 90, 120, and 180 min was estimated. The hydrolysis index (HI) was calculated from the ratio between the area under the hydrolysis curve compared with a reference sample (white bread). The pGI was estimated from the HI, and relative values were calculated using the following equation (Equation (6)).
pGI = 39.71 + 0.549 (HI)(6)

### 2.9. Transmittance and Wavelength of Maximum Absorption

Gel clarity was assessed by the method suggested by de la Rosa-Millan [10]. Briefly, a 1% de-ionized water dispersion of gelatinized starch was prepared, cooled at room temperature, then poured into spectrophotometer cells and read in transmittance mode (%T) at 650 nm against a distilled water blank (whose %T = 100%). To estimate the extent of changes in linear structures, the wavelength of Maximum Absorption (λmax) was calculated by quantifying the formation of iodine complexes of starches with an iodine solution (2% KI, 0.2% I_2_) according to the protocol described by Tetchi, Rolland-Sabaté, Amani, and Colonna [13] using a Genesys 2000 spectrophotometer (Thermo Fisher Scientific, Waltham, MA, USA) set to record absorbances in the range between 450 and 700 nm.

### 2.10. Emulsifying Characteristics

The method described by Pedroche, Yust, Lqari, Girón-Calle, Alaiz, Vioque, and Millán [14] with slight modifications was used. Briefly, 1 g of the sample was mixed with 25 mL de-ionized water and homogenized with an Ultra Turrax device (Mod. T-18, Ika Werke, Germany, at 10,000 rpm for 1 min at room temperature (≈25 °C), and their pH was adjusted to 7.0 (either with 0.1 M HCl or 0.1 M NaOH). After, the solution was mixed with 25 mL of soybean oil, followed by mechanical homogenization at 10,000 rpm for 1 min in an Ultra Turrax. Finally, the emulsion was centrifuged at 1300× *g* for 5 min and calculated according to Equation (7).
(7)$$EA\;\left(\%\right)=\frac{Height\;of\;emulsified\;layer}{Total\;height\;of\;the\;contents}\times 100$$

Emulsion stability (*ES*) was measured in the previously emulsified tubes heated at 80 °C in a water bath for 30 min, followed by centrifugation at 1300× *g* for 5 min. Their heat stability was calculated as follows (Equation (8)):(8)$$ES\;\left(\%\right)=\frac{Height\;of\;remaining\;emulsion\;layer}{Height\;of\;original\;emulsified\;layer}\times 100$$

### 2.11. Foaming Characteristics

Foaming capacity (*FC*) was estimated in a dispersion of 500 mg of starch in 50 mL of de-ionized water. The pH of the dispersion was adjusted to pH 7.0 with either 1M NaOH or 1N HCL, followed by an Ultra Turrax homogenizer (Mod. T-18, Ika Werke, Germany) at 10,000 rpm for 2 min. After the whipped solution was transferred into a 100 mL graduated cylinder, and their volume was recorded and compared with a similar dispersion without stirring, *FC* was expressed as the increased volume (%) due to stirring, using the following equation (Equation (9)):(9)$$FC\;\left(\%\right)=\frac{volume\;after\;whipping\;\left(\mathrm{mL}\right)-volume\;before\;whipping\;\left(\mathrm{mL}\right)}{volume\;before\;whipping\;\left(\mathrm{mL}\right)}\times 100$$

The foam stability (*FS*) was calculated as the difference in volume (%) of the whipped starch dispersion after 30 min of standing (at ≈ 25 °C). The foam stability was then calculated using the following equation (Equation (10)).
(10)$$FS\;\left(\%\right)=\frac{volume\;after\;whipping\;\left(\mathrm{mL}\right)-volume\;after\;30\;\min\;\left(\mathrm{mL}\right)}{volume\;after\;whipping\;\left(\mathrm{mL}\right)}\times 100$$

### 2.12. Molecular Characteristics of Starch and Amylopectin Chain Length Distribution

Samples were dissolved in 90% DMSO according to the methodology of Yoo and Jane [15]. A total of 500 mg of starch powder was dispersed in 10 mL of 90% DMSO solution. The suspension was mechanically stirred for 1 h in a boiling water bath and was kept stirring for 24 h at 25 °C. An aliquot (2 mL) of the dispersion was mixed with four volumes of ethanol (8 mL) to precipitate starch. The insoluble starch fraction was separated by centrifugation at 7000 rpm for 20 min and redissolved in 5 mL of hot water (to obtain the concentration of 4 mg/mL), and finally stirred for 30 min in a boiling water bath. The samples were filtered through a 5.0 µm pore size nylon membrane for chromatographic analysis, in all cases no visible residues were attached to the syringe filters.

The weight-average molar mass and z-average gyration of amylopectin were determined using high-performance size-exclusion chromatography equipped with multi-angle laser-light scattering and refractive index detectors (HPSEC-MALLS-RI). Starch samples were prepared as described by Yoo and Jane [15]. The HPSEC system consisted of a Waters 1525 binary pump (Waters Corp, Milford, MA, USA), a multi-angle laser-light scattering detector (Dawn, Wyatt Tech. Co., Santa Barbara, CA, USA), and a Waters 2414 refractive index detector. A Shodex OH Pak KB-guard column and an SB-806 HQ column (Showa Denko KK, JM Science, Grand Island, NY, USA) were used to separate amylopectin from amylose. Operating conditions and data analysis were similar to those described by Yoo and Jane [15], except the flow rate used was 0.5 mL/min, and the sample concentration was 4 mg/mL.

The debranching of the starch was carried out according to the procedure described by Ao, Simsek, Zhang, Venkatachalam, Reuhs, and Hamaker [16], using 15U of isoamylase (E-ISAMY, Megazyme, Wicklow, Ireland), followed a by 6 h incubation under constant stirring at 45 °C. The enzyme was inactivated by boiling the samples in a water bath for 20 min. Immediately after debranching, the solution was filtered on a 0.45 µm pore size nylon membrane and injected into a CLARET-IR system consisting of a Waters 1525 binary pump (Waters Corp, Milford, MA, USA), a Waters 2414 refractive index detector, and two HR 10/30 columns connected simultaneously the first containing Superdex 200. The second was with Superdex 30 gel (Amersham Biosciences, Piscataway, NJ, USA), with a 0.4 mL/min flow rate, using de-ionized water (18 MΩ). Pullulan standards with known molecular weight were used (180, 738, 5900, 11,800, 22,800, 47,300 and 112,000 g/mol) (Agilent, Santa Clara, CA, USA). Each sample was analyzed in duplicate using the Empower 3 software (Waters Corp.).

### 2.13. Statistical Analyses

All experiments and procedures described in this research were performed in triplicate unless otherwise specified. A one-way variance analysis was performed, and when significant differences were found at the significance level of 0.05, a Tukey test of multiple comparisons was used. Additionally, a Principal component analysis was performed to explain the effect of the molecular composition and interactions. All data were analyzed using the Minitab software (Ver. 19.2020, Minitab, Chicago, IL, USA).

## 3. Results and Discussion

### 3.1. Starch Characteristics

The addition of SSL promoted a dilution effect on the starch content, but no significant differences were detected when extrusion was used. (Table 3). This behavior has been previously reported since thermal processes as extrusion did not impact starch composition but often promoted changes at a molecular level [7,17]. The harsh shear conditions may damage or dextrinize their fractions (amylose and amylopectin). The amylose content decreased from ≈27% to ≈21% in maize starch and from ≈21% to ≈17% in sorghum due to the thermo-mechanical process. Consequently, the amylopectin content (calculated by difference) showed an increase that could be related to partial on-site complexation of resulting amylose dextrins with amylopectin molecules, decreasing the soluble fraction used to determine amylose content. This was observed as large aggregates in the micrographs shown in Figure 1. Other studies have shown that the variations in the amylose-amylopectin ratio depend on granular degradation and glucan chain dextrinization related to the extrusion thermomechanical process, which affects the function and digestion properties of the resulting products [18].

### 3.2. Thermal Characteristics

When thermal treatments, such as extrusion, are applied to starches, several changes in the crystalline structure occur, affecting their granular characteristics and functionality. Such changes often result in improved water absorption due to newly exposed –OH groups and the partial or total disruption of crystallites due to the depolymerization of amylose chains and amylopectin clusters, evidenced by lower thermal transition intervals (Tf-To) and lower viscosity [19]. For this reason, assessing the thermal characteristics of treated starches is necessary since many of them (according to their specific application) may endure additional thermal processes. Both native starches showed similar thermal profiles with slight variations in gelatinization temperature parameters; furthermore, they showed similar values in their gelatinization intervals and significant differences in their enthalpy values (ΔH), founding 12.43 J/g in maize and 14.21 J/g in sorghum (Table 4). The different processing conditions on both starches significantly affected these parameters, most evident in the enthalpy values around 4 J/g reduced in all cases. The observed decrease is the consequence of the loss of the crystalline structure promoted by the thermo-mechanical process; this is reflected in the partial gelatinization of the extrudates that occurred to a 60% to 70% extent in the case of sorghum. Additionally, it is important to understand that the prevalence of crystallites influenced some of their physicochemical and functional properties; for this, we evaluated their crystallinity percentage, which gives information related to their molecular arrangement. In the case of maize, the crystallinity was 27%, and in sorghum, around 29%. The use of SSL alone influenced these values; in the case of maize starch, it decreased to 25.36%, and for sorghum to 27.15%; these reductions could be influenced by the addition of this component that may promote partial instability due to the different heat capacity. Additionally, the mechanical and thermal extrusion process significantly affected such values [20].

### 3.3. In Vitro Starch Digestion Properties

We evaluated the starch digestion fractions to understand how SSL and extrusion conditions impact their digestion. The RDS fraction significantly increased due to the extrusion conditions, from 60% to 75% for maize and from 62% to 72% for sorghum. The SDS fraction decreased from 22.27% to ≈18% in native maize. Meanwhile, the RS fraction increased from 1.2% to 21.41%; this variability could be related to the changes in the molecular structure where molecular weight, starch chain length, and amylopectin crystalline polymorphism influenced this behavior [10,16]. In sorghum starch, similar results on RDS and SDS were found; however, the RS showed high variability from 9.4 to 0.13% (Table 5). These similarities could be related to the amylopectin type where a higher proportion of short chains and granular characteristics as size and shape may promote a similar behavior during extrusion in both starches; SSL does not seem to affect their performance [18,21] significantly.

### 3.4. Functional Properties

Among the different uses of starches in food formulations, clarity, emulsifying properties, and foaming capacity are commonly considered relevant since these are of industrial importance for various applications [22]. For discussion purposes, only selected treatments (native, native + SSL and two processing conditions will be discussed in this section), the native starches showed a transmittance value of 26.36% for maize and 29.41% for sorghum, which decreased during the storage time (7 days) to reach values ≈5% in both cases. SSL did not significantly affect this parameter; however, maize starch improved to reach 91.1% when extruded at 100 rpm, 40% moisture, and 80 °C. On sorghum, different processing conditions promote an increase in gel clarity, reaching 69.31%, but after storage, the values were similar to non-extruded samples (Figure 2A). Regarding emulsion properties, they are one of the food applications that often require chemical derivatization on starches to achieve the desired results [7,17]. Both native starches showed a weak emulsion capacity, even when SSL was used. Still, it increased significantly (*p* < 0.05) after extrusion and was further enhanced with SSL (Figure 2B). As in the previous analysis, the foaming capacity improved when SSL and extrusion were combined, reaching values of 83.7% with 100 rpm, 40% moisture, and 80 °C with SSL. Such enhancements may be promoted due to the increase in small Mw components, such as dextrins and small linear glucans generated during the extrusion, that have been shown to enhance complexing with each other and with oil and other molecules, improving the stability of the gel and emulsions, respectively (Figure 2C) [21,23].

### 3.5. Molecular Characteristics and Chain Length Distribution

Previous studies have found it challenging to associate the influence of remaining molecular structures from starch particles after a dextrination or mechanical breakdown process; since most of the attributed functional and digestion behavior is related to its fine structure [17,18,24]. When analyzing these characteristics, we realize that the amylopectin fraction on the chromatograms showed a broad distribution (Appendix A) but lower Mw, which is online with the results from the enzymatic test; the differences in the chromatograms were slight but evident and would correspond to degraded Mw fractions that could not be efficiently determined by the enzymatic method. In other studies, the presence of this fraction has been related to non-conventional behavior in pasting and thermal characteristics and could influence their digestion profile and some of their functional properties [25]. The amylopectin Mw were 3.40 and 3.33 × 10^8^ g/mol, with an Rz of 323 and 298 nm for native maize and sorghum, respectively, which confirmed the molecular similarities between these two starch sources. The addition of SSL did not affect this value (*p* < 0.05) (Figure 3A). Other studies have related the amylopectin molecular structure with different behaviors affecting the functional properties, particularly thermal stability, which is often enhanced when large and organized crystals are present [26]. Under extrusion, average Mw decreased up to 2 and 1.64 × 10^8^ g/mol for maize and sorghum, Rz also decreased a large extent, founding sizes from 73.47 up to 199.37 nm in size in maize and from 75.58 to 105.86 nm for sorghum (Figure 3B). Regarding the amylose fraction, we found a significant decrease in the chromatogram areas when extrusion was applied and a noticeable shift from larger chains to small chains (≈1500 to ≈500 DP), which is more evident in sorghum starch. This behavior shows that the sorghum molecular structure could be more susceptible to mechanical damage. In other studies, similar results were found when cereal starches were processed under mechanical treatments, finding that the crystallite organization is prone to receive most of the mechanical energy. At the same time, the amylose fraction presents less damage, as others suggest [24,27]. Due to their potential application as a food ingredient and in order to understand their functionality mechanisms, these starches were debranched to reveal their fine structure [15]. In both sources, the B1 chains (13–24 DP) were prevalent, being medium size chains commonly located at the amylopectin clusters exterior; when extrusion was applied, the A-chains (6–12 DP) significantly increased their proportion while decreasing the B2 (25–36 DP) and B3 (>36 DP) chains, which would correspond to the amylopectin mainframe, that supports the other chain types (Figure 3C). These results demonstrate that extrusion promoted extensive dextrinization or depolymerization of larger glucan chains in the amylose fraction. This leads us to think that most mechanical damage during extrusion promoted the rupture of amylopectin clusters rather than amylose [7,17,23].

### 3.6. Statistical Analysis

PCA analyses helped understand the extent of molecular interactions within samples, which otherwise would be complicated to explain. We choose the variables related to the starch architecture, in vitro digestibility, and functional properties. The first and second components (PC1 and PC2) accounted for an accumulative variance of 85.4%. The score plot showed significant differences when extrusion with and without SSL was tested, differentiating between starch sources (Figure 4A). The loading plot (Figure 4B) showed the correlations among the evaluated variables. The main contributor factors were the crystalline structure assessed by X-ray diffraction, influenced by amylopectin B3 type chains. The second component was the RS fraction, followed by B2-type chains of amylopectin, which other authors have related to improved emulsion stability [4,6]. That could stabilize the system. These results showed that the cluster arrangement and amylopectin fine architecture of these two starch sources preserve many of the granules’ structural characteristics (as seen in Figure 1) and, at the same time, expose its –OH groups, promoting enhanced interactions when used as emulsifiers or foam promoting ingredients, mainly when SSL was used in the process.

## 4. Conclusions

Both cereal starches presented similar crystalline characteristics that changed after extrusion. The use of SSL impacts their starch functionality as emulsifiers, foam promoters, and stabilizers by improving their molecular interactions. The influence of amylopectin’s fine molecular structure was relevant since linear glucans, particularly B2 and B3 chains, helped to stabilize the emulsions and generate stable foams, possibly by a molecular interaction mechanism. Enzyme hydrolysis was affected by the molecular composition of starches, promoting differences in the digestion rate that was correlated with the amylopectin crystalline structure, particularly B3 type chains, as well as the resulting long glucans related to amylose. This research’s outcome and potential applications would be useful for developing new food ingredients with broad applications without chemical derivatization.

## Figures and Tables

**Figure 1 foods-12-01988-f001:**
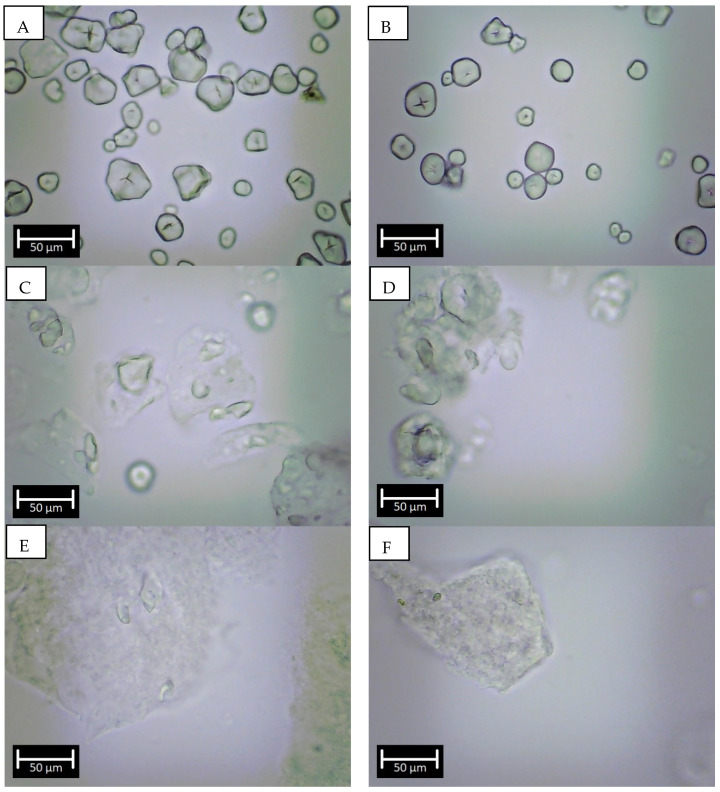
Morphology observations of (**A**) Native maize, (**B**) Native sorghum, (**C**) Extruded maize, (**D**) Extruded sorghum, (**E**) Extruded maize with + SSL, and (**F**) Extruded sorghum + SSL. SSL: Sodium Stearoyl Lactylate.

**Figure 2 foods-12-01988-f002:**
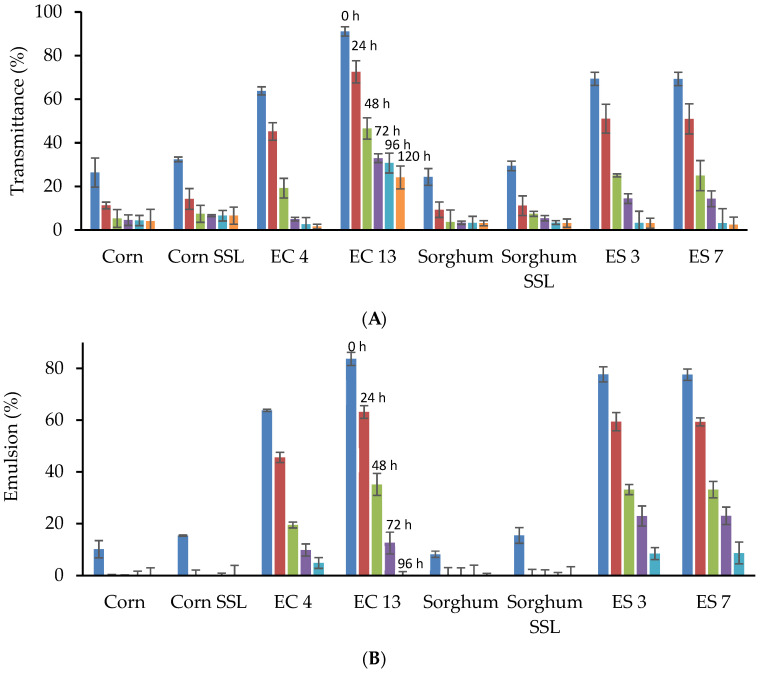
Functional properties of native and selected extruded samples. (**A**) Gel clarity, (**B**) Emulsion capacity, and (**C**) Foam capacity. SSL: Sodium Stearoyl Lactylate, EC: Extruded corn, and ES: Extruded sorghum.

**Figure 3 foods-12-01988-f003:**
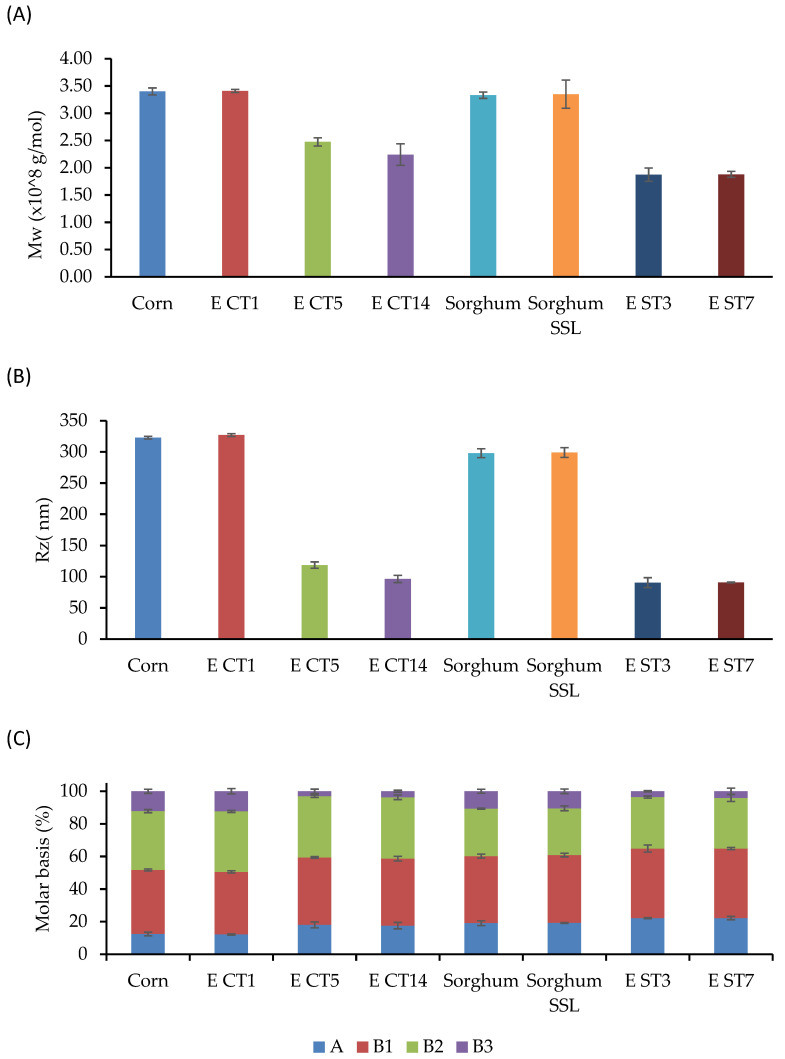
Molecular changes in native and selected extruded starches. (**A**) Molecular weight, (**B**) Hydrodynamic radius (Rz), and (**C**) Chain length distribution.

**Figure 4 foods-12-01988-f004:**
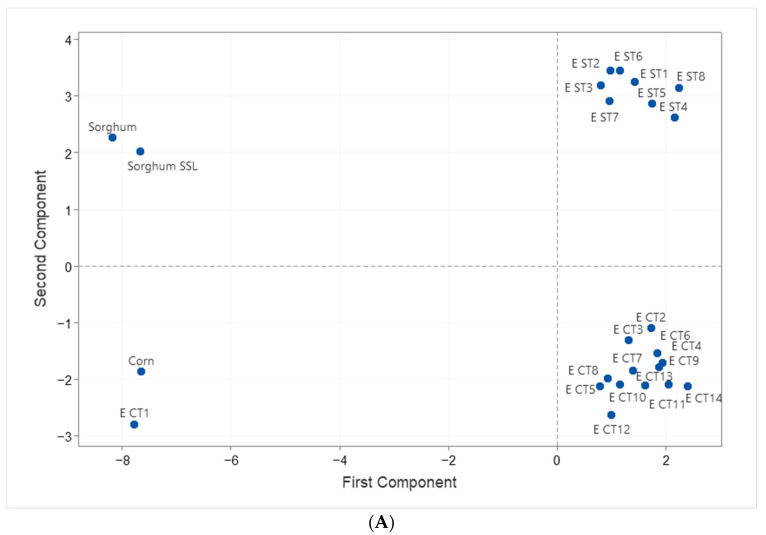
Meaningful correlation analysis of native and extruded starches added with SSL. (**A**) Score plot, (**B**) Loading plot. SSL: Sodium Stearoyl Lactylate.

**Table 1 foods-12-01988-t001:** Box–Behnken design to determine optimal extrusion conditions for maize starch.

Treatment	RPM	Moisture (%)	Temp (°C)	SSL (%)
E CT1	140	30	60	0
E CT 2	100	30	80	0
E CT 3	180	30	80	0
E CT 4	140	20	80	0
E CT 5	140	40	80	0
E CT 6	140	30	100	0
E CT 7	140	20	100	0.5
E CT 8	140	40	100	0.5
E CT 9	100	30	100	0.5
E CT 10	180	30	100	0.5
E CT 11	100	20	80	0.5
E CT 12	180	20	80	0.5
E CT 13	100	40	80	0.5

E CT: Extruded Corn Treatments, RPM: Revolution per minute, SSL: Sodium Stearoyl Lactylate.

**Table 2 foods-12-01988-t002:** Factorial (23) design to determine the optimal conditions to extrude sorghum starch.

Treatment	SSL (%)	Temp (°C)	Moisture (%)
E ST1	0.5	80	40
E ST2	0.5	80	20
E ST3	0.5	100	20
E ST4	0.5	100	40
E ST5	1	80	40
E ST6	1	80	20
E ST7	1	100	20
E ST8	1	100	40

E ST: Extruded sorghum treatments, RPM: Revolution per minute, SSL: Sodium Stearoyl Lactylate, and Temp: Temperature.

**Table 3 foods-12-01988-t003:** Effect of SSL addition and extrusion conditions evaluated on maize and sorghum starches amylose and amylopectin content.

Treatment	Total Starch (%)	Amylose (%)	Amylopectin (%)
Maize	91.23 ± 0.41 b	27.35 ± 0.15 a	72.65 ± 0.42 g
E CT1	88.54 ± 0.30 c	25.65 ± 0.50 b	74.35 ± 0.29 f
E CT2	89.42 ± 0.52 c	23.24 ± 0.12 c	76.76 ± 0.17 d
E CT3	90.07 ± 0.73 b	24.09 ± 0.77 b	75.91 ± 0.30 e
E CT4	88.99 ± 0.16 c	22.21 ± 0.39 d	77.79 ± 0.81 c
E CT5	89.61 ± 0.40 c	23.33 ± 0.90 c	76.67 ± 0.53 d
E CT6	90.22 ± 0.24 b	23.74 ± 0.03 c	76.26 ± 0.03 d
E CT7	89.88 ± 0.03 c	23.30 ± 0.60 c	76.70 ± 0.58 d
E CT8	90.05 ± 0.52 b	23.57 ± 0.69 c	76.44 ± 0.18 d
E CT9	90.13 ± 0.09 b	23.45 ± 0.12 c	76.55 ± 0.05 d
E CT10	90.00 ± 0.06 b	23.82 ± 0.49 c	76.19 ± 0.63 d
E CT11	88.42 ± 0.85 c	21.43 ± 0.99 d	78.57 ± 0.53 c
E CT12	88.59 ± 0.50 c	22.70 ± 0.09 d	77.30 ± 0.78 c
E CT13	89.09 ± 0.17 c	22.41 ± 0.86 d	77.59 ± 0.54 c
E CT14	88.81 ± 0.65 c	22.73 ± 0.36 d	77.27 ± 0.84 c
Sorghum	93.23 ± 0.22 a	21.23 ± 0.22 d	78.77 ± 0.10 b
Sorghum SSL	91.93 ± 0.62 b	19.92 ± 0.05 e	80.08 ± 0.20 b
E ST1	91.19 ± 0.40 b	17.98 ± 0.56 f	82.02 ± 0.67 a
E ST2	92.00 ± 0.58 b	19.00 ± 0.51 e	81.00 ± 0.94 a
E ST3	91.81 ± 0.75 b	18.60 ± 0.11 e	81.40 ± 0.20 a
E ST4	90.50 ± 0.31 b	17.09 ± 0.02 f	82.91 ± 0.79 a
E ST5	91.82 ± 0.38 b	18.61 ± 0.99 e	81.39 ± 0.79 b
E ST6	92.13 ± 0.16 b	19.12 ± 0.40 e	80.88 ± 0.03 b
E ST7	90.93 ± 0.54 b	17.72 ± 0.83 f	82.28 ± 0.48 a
E ST8	90.54 ± 0.69 b	17.13 ± 0.43 f	82.87 ± 0.50 a

E CT: Extruded Corn Treatments, E ST: Extruded sorghum treatments, RPM: Revolution per minute, SSL: Sodium Stearoyl Lactylate. The results are the average of three replicates ± std deviation. Different letters in the same column reflect significant differences (*p* < 0.05).

**Table 4 foods-12-01988-t004:** Effect of SSL addition and extrusion conditions evaluated on maize and sorghum starches on thermal properties.

Treatment	To (°C)	Tp (°C)	Tc (°C)	Tf-To (°C)	ΔH (J/g)	%Gel	Crystallinity (%)
Maize	66.43 ± 1.81 a	70.31 ± 1.81 b	73.26 ± 1.81 b	6.83 ± 1.81 c	12.43 ± 1.81 b	ND	27.41 ± 1.81 a
E CT1	64.93 ± 0.51 b	68.81 ± 0.51 c	69.13 ± 0.51 c	4.20 ± 0.51 d	10.93 ± 0.51 c	12.07 ± 0.51 d	25.36 ± 0.51 b
E CT2	55.32 ± 0.94 d	57.41 ± 0.94 e	59.36 ± 0.94 e	4.04 ± 0.94 d	5.36 ± 0.94 d	56.88 ± 0.94 c	11.26 ± 0.94 d
E CT3	54.62 ± 2.46 d	56.71 ± 2.46 e	60.24 ± 2.46 e	5.62 ± 2.46 c	4.85 ± 2.46 d	60.97 ± 2.46 b	10.51 ± 2.46 d
E CT4	53.65 ± 2.42 d	55.74 ± 2.42 e	59.87 ± 2.42 e	6.22 ± 2.42 c	4.66 ± 2.42 d	62.52 ± 2.42 b	11.42 ± 2.42 d
E CT5	52.85 ± 2.23 d	54.94 ± 2.23 e	58.69 ± 2.23 e	5.84 ± 2.23 c	4.78 ± 2.23 d	61.55 ± 2.23 b	12.29 ± 2.23 d
E CT6	51.99 ± 2.28 d	54.08 ± 2.28 e	57.98 ± 2.28 e	5.99 ± 2.28 c	4.73 ± 2.28 d	61.94 ± 2.28 b	12.19 ± 2.28 d
E CT7	51.09 ± 1.53 d	53.18 ± 1.53 f	57.15 ± 1.53 e	6.07 ± 1.53 c	4.71 ± 1.53 d	62.13 ± 1.53 b	11.84 ± 1.53 d
E CT8	50.22 ± 0.29 e	52.31 ± 0.29 f	56.21 ± 0.29 f	5.99 ± 0.29 c	4.73 ± 0.29 d	61.95 ± 0.29 b	12.33 ± 0.29 d
E CT9	49.28 ± 1.76 e	51.37 ± 1.76 f	55.43 ± 1.76 f	6.14 ± 1.76 c	4.68 ± 1.76 d	62.34 ± 1.76 b	8.81 ± 1.76 e
E CT10	48.52 ± 1.94 e	50.61 ± 1.94 f	54.28 ± 1.94 f	5.77 ± 1.94 c	4.80 ± 1.94 d	61.36 ± 1.94 b	12.89 ± 1.94 d
E CT11	47.48 ± 0.32 e	49.57 ± 0.32 g	53.85 ± 0.32 f	6.37 ± 0.32 c	4.61 ± 0.32 d	62.92 ± 0.32 b	12.04 ± 0.32 d
E CT12	46.81 ± 2.19 e	48.90 ± 2.19 g	52.36 ± 2.19 f	5.54 ± 2.19 c	4.87 ± 2.19 d	60.78 ± 2.19 b	13.11 ± 2.19 d
E CT13	45.88 ± 0.85 e	47.97 ± 0.85 g	52.02 ± 0.85 f	6.14 ± 0.85 c	4.68 ± 0.85 d	62.34 ± 0.85 b	8.81 ± 0.85 e
E CT14	45.14 ± 0.31 e	47.23 ± 0.31 g	50.84 ± 0.31 g	5.69 ± 0.31 c	4.83 ± 0.31 d	61.17 ± 0.31 b	7.63 ± 0.31 e
Sorghum	69.43 ± 1.87 a	74.32 ± 1.87 a	76.26 ± 1.87 a	6.83 ± 1.87 c	14.21 ± 1.87 a	ND	29.34 ± 1.87 a
Sorghum SSL	67.93 ± 1.46 a	72.82 ± 1.46 a	73.13 ± 1.46 b	5.20 ± 1.46 c	12.71 ± 1.46 b	10.56 ± 1.46 d	27.15 ± 1.46 a
E ST1	57.25 ± 0.20 c	62.14 ± 0.20 d	64.15 ± 0.20 d	6.90 ± 0.20 c	5.36 ± 0.20 d	62.28 ± 0.20 b	13.17 ± 0.20 d
E ST2	49.72 ± 1.01 e	54.61 ± 1.01 e	57.21 ± 1.01 e	7.49 ± 1.01 b	4.39 ± 1.01 d	69.08 ± 1.01 a	17.41 ± 1.01 c
E ST3	50.52 ± 1.87 d	55.41 ± 1.87 e	60.15 ± 1.87 e	9.63 ± 1.87 a	4.20 ± 1.87 d	70.43 ± 1.87 a	13.17 ± 1.87 d
E ST4	48.85 ± 0.98 e	53.74 ± 1.98 e	58.43 ± 1.98 e	9.58 ± 1.98 a	4.01 ± 1.98 d	71.79 ± 1.98 a	7.31 ± 1.98 e
E ST5	47.24 ± 0.43 e	52.13 ± 0.43 e	54.13 ± 0.43 f	6.89 ± 0.43 c	4.20 ± 0.43 d	70.47 ± 0.43 a	13.03 ± 0.43 d
E ST6	48.08 ± 1.51 e	52.97 ± 1.51 e	55.26 ± 1.51 f	7.18 ± 1.51 b	4.39 ± 1.51 d	69.11 ± 1.51 a	14.16 ± 1.51 d
E ST7	48.87 ± 1.46 e	53.76 ± 1.46 e	55.12 ± 1.46 f	6.25 ± 1.46 c	4.20 ± 1.46 d	70.47 ± 1.46 a	13.03 ± 1.46 d
E ST8	47.20 ± 0.50 e	52.09 ± 0.50 e	54.33 ± 0.50 f	7.13 ± 0.50 b	4.00 ± 0.50 d	71.82 ± 0.50 a	7.16 ± 0.50 e

E CT: Extruded Corn Treatments, E ST: Extruded sorghum treatments, SSL: Sodium Stearoyl Lactylate, To: Onset gelatinization temperature, Tp: Peak gelatinization temperature, Tc: Conclusion gelatinization temperature, ΔH: Gelatinization enthalpy, and %Gel: Gelatinization percentage. The results are the average of three replicates ± std deviation. Different letters in the same column reflect significant differences (*p* < 0.05).

**Table 5 foods-12-01988-t005:** Effect of SSL addition and extrusion conditions evaluated on maize and sorghum starches on starch digestion fractions. Different letters in the same columns mean.

Treatment	RDS (%)	SDS (%)	RS (%)	pGI
Maize	73.43 ± 1.83 b	22.27 ± 0.18 b	4.30 ± 0.69 f	92.18 ± 1.25 a
E CT1	64.91 ± 2.87 c	24.43 ± 3.57 b	10.66 ± 0.46 e	90.41 ± 3.61 a
E CT2	72.24 ± 2.65 b	20.52 ± 3.88 b	7.24 ± 4.15 e	93.18 ± 3.10 a
E CT3	68.90 ± 1.40 c	20.19 ± 4.22 b	10.91 ± 4.23 e	92.80 ± 0.57 a
E CT4	75.26 ± 2.83 b	19.78 ± 2.66 b	4.96 ± 4.26 f	93.66 ± 2.43 a
E CT5	66.91 ± 3.44 c	18.94 ± 2.79 b	14.15 ± 0.46 d	92.71 ± 2.43 a
E CT6	70.25 ± 1.60 b	19.28 ± 1.69 b	10.48 ± 3.30 e	93.09 ± 3.70 a
E CT7	71.92 ± 3.56 b	19.44 ± 3.40 b	8.64 ± 3.00 e	93.28 ± 2.30 a
E CT8	70.33 ± 1.90 b	19.28 ± 3.89 b	10.38 ± 2.21 e	93.10 ± 4.19 a
E CT9	73.67 ± 0.52 b	19.62 ± 3.68 b	6.71 ± 0.12 f	93.48 ± 2.29 a
E CT10	65.32 ± 1.57 d	18.78 ± 0.51 c	15.90 ± 1.11 c	92.53 ± 3.80 a
E CT11	78.68 ± 1.07 a	20.12 ± 3.77 b	1.20 ± 3.99 g	94.05 ± 2.59 a
E CT12	60.31 ± 3.29 e	18.28 ± 3.15 c	21.41 ± 3.98 a	91.95 ± 1.59 a
E CT13	73.67 ± 0.17 b	19.62 ± 2.08 b	6.71 ± 1.88 f	93.48 ± 0.87 a
E CT14	63.65 ± 4.05 d	18.61 ± 4.12 c	17.73 ± 0.24 b	92.34 ± 4.17 a
Sorghum	62.71 ± 3.90 d	27.88 ± 0.15 a	9.41 ± 3.35 e	90.37 ± 2.03 a
Sorghum SSL	62.76 ± 0.13 d	27.88 ± 2.03 a	9.37 ± 1.74 e	90.15 ± 3.29 a
E ST1	71.73 ± 4.00 b	27.12 ± 0.64 a	1.15 ± 0.89 g	93.00 ± 2.10 a
E ST2	72.11 ± 1.68 b	27.71 ± 0.60 a	0.18 ± 2.10 g	92.94 ± 4.13 a
E ST3	70.63 ± 3.27 b	26.85 ± 1.95 a	2.52 ± 0.80 g	92.89 ± 1.53 a
E ST4	65.53 ± 3.49 d	25.07 ± 2.10 a	9.40 ± 1.34 e	92.51 ± 0.85 a
E ST5	65.73 ± 3.85 d	25.61 ± 3.82 a	8.66 ± 1.36 e	92.43 ± 4.02 a
E ST6	72.15 ± 1.70 b	27.71 ± 1.31 a	0.13 ± 0.01 h	92.94 ± 2.39 a
E ST7	69.44 ± 1.20 c	26.54 ± 1.53 a	4.03 ± 1.66 f	92.78 ± 3.26 a
E ST8	72.27 ± 3.96 b	26.74 ± 1.71 a	0.99 ± 0.12 g	93.15 ± 0.27 a

E CT: Extruded Corn Treatments, E ST: Extruded sorghum treatments, SSL: Sodium Stearoyl Lactylate, RDS: Rapidly digestible starch, SDS: Slowly digestible starch, RS: Resistant starch, and pGI: Predicted Glycemic Index. The results are the average of three replicates ± std deviation. Different letters in the same column reflect significant differences (*p* < 0.05).

## Data Availability

The data presented in this study are available upon request from the corresponding author.

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
