# Peer review of "Physicochemical and In Vitro Starch Residual Digestion Structures of Extruded Maize and Sorghum Starches Added with Sodium Stearoyl Lactylate"

_foods, 2023, doi:10.3390/foods12101988_

Round 1
Reviewer 1 Report
In the manuscript entitled “Physicochemical and in vitro starch residual digestion structures of extruded maize and sorghum starches added with sodium stearoyl lactylate” presented by Julian de la Rosa Millan et al., studied the physicochemical, in vitro digestion, and the structural features digestion residues of maize and sorghum starches subjected to thermoplastic extrusion, along with the influence of Sodium Stearoyl Lactylate (SSL) and they presented an extensive characterization.
The work is quite consequent. The work would help to understand the effect of the thermoplastic extrusion on starch molecules and their functionalities.
However, there are some issues which must be revised and explained before consideration for publication in Foods.
Authors are encouraged to use the Microsoft Word template proposed by mdpi.
All tables should be revised and prepared according to guides for authors (police, size, formats etc…)
subtitles need to be standardized as mentioned in guides for authors (see Microsoft word template).
Make space before °C and units as mentioned in marked pdf.
Table 1 and 2: why did authors use 2 different design experiments. Studies could be conducted using box-Behnken design on maize starch and the optimum can then be used for validation on sorghum starch.
Section 2.12. Any idea about recovery after filtration through nylon membrane and then also for SEC-HPLC?
Section 3.1. contradiction, in table 3, we can see significant differences between samples for starch content!!!!!?
Line 269-285: how do authors explain the decrease of AM content!!! The discussion should be developed! It is about enzymatic determination!!! Content should be the same!!!!?
Section 3.5. using SEC-MALS, authors have an opportunity to check results of table 3. Check amylose and amylopectin content!!!? SEC-HPLC-RI results corroborate megazyme results?
Authors can use SEC-HPLC-RI chromatogram for comparison!?
There is no degradation of amylose and amylopectin producing a very small molecules that we can find in chromatogram!!!?
Discussion should be improved!

Author Response
In the manuscript entitled “Physicochemical and in vitro starch residual digestion structures of extruded maize and sorghum starches added with sodium stearoyl lactylate” presented by Julian de la Rosa Millan et al., studied the physicochemical, in vitro digestion, and the structural features digestion residues of maize and sorghum starches subjected to thermoplastic extrusion, along with the influence of Sodium Stearoyl Lactylate (SSL) and they presented an extensive characterization.
The work is quite consequent. The work would help to understand the effect of the thermoplastic extrusion on starch molecules and their functionalities.
However, there are some issues which must be revised and explained before consideration for publication in Foods.
Authors are encouraged to use the Microsoft Word template proposed by mdpi.
All tables should be revised and prepared according to guides for authors (police, size, formats etc…)
Subtitles need to be standardized as mentioned in guides for authors (see Microsoft word template).
Make space before °C and units as mentioned in marked pdf.
The format was revised and corrections were made thorough the text.
Table 1 and 2: why did authors use 2 different design experiments. Studies could be conducted using box-Behnken design on maize starch and the optimum can then be used for validation on sorghum starch.
R= We appreciate your time to review this work. Even when maize starch and sorghum starch could be very similar in molecular structure we considered to extend the study on sorghum to validate if could be any difference during extrusion. A sentence justifying this approach was inserted in the reviewed version of the manuscript. See lines 102-104.
Section 2.12. Any idea about recovery after filtration through nylon membrane and then also for SEC-HPLC?
R= In this case we did not estimate the recovery, nor we notice that clumps or hard to dissolve structures were present. Also, we did not recover the fractions (although this would be interesting). To clarify this, a sentence about sample preparation is included in the revised fraction. See lines 210-211 and 217-218.
Section 3.1. contradiction, in table 3, we can see significant differences between samples for starch content!!!!!?
R= We appreciate the observation. The data was revised and there were errors in the assignation of the letter, now is corrected and the discussion regarding this table was edited.
Line 269-285: how do authors explain the decrease of AM content!!! The discussion should be developed! It is about enzymatic determination!!! Content should be the same!!!!?
R= The sentences related to the decrease of amylose fraction were updated. We hypothesize that the extrusion process promoted partial dextrination of both starch molecules but at the same time promoted physical molecular interaction (on-site complexation), which was also related to decreased solubility of such samples. This also observed in Figure 1, were larger starch clumps were observed under the microscope, even larger than native starch granules. See lines 251-261.
Section 3.5. using SEC-MALS, authors have an opportunity to check the results of table 3. Check amylose and amylopectin content!!!? SEC-HPLC-RI results corroborate megazyme results?
Authors can use SEC-HPLC-RI chromatogram for comparison!?
There is no degradation of amylose and amylopectin producing very small molecules that we can find in the chromatogram!!!?
Discussion should be improved!
R= The recommendations of the referee were considered. A supplementary figure with selected HPSEC data is included. The discussion section about molecular structure characteristics is updated. See lines 347-351 and Figure S1.
Reviewer 2 Report
line 32 "wet-milling" - maybe it would be better to generalize and replace it with the term "grinding" because in the case of potato tubers it is abrasion, but in cereals it is actually gri
line 41 - "formulations " isn't "structuring" better?
in order to work lines 66- SSL should be in brackets and the full name before it
2.2. Starch extrusion - no such term "Regular maize" - native?
line 75 "...were kindly provided by.." - there is a separate place in the publication for acknowledgments. There should only be statements here.
2.4. Starch and amylose contents - description too short, please extend the methodology
line 159- "against a distilled water blank" worth adding, whose T%=100..in brackets
Very well described methodology regarding functional properties (emulsifiability and foam capacity and stability)
line 417"starch content" I don't understand, it's not a conclusion??????Please explain
In my opinion, the conclusions are too short and prove the purpose of the research undertaken.
Author Response
Line 32 "wet-milling" - maybe it would be better to generalize and replace it with the term "grinding" because in the case of potato tubers it is abrasion, but in cereals it is actually gri
R= We appreciate the observation of the referee was taken into account. Sentence was removed and introduction updated.
line 41 - "formulations " isn't "structuring" better?
R= Observation of the referee was taken into account. Sentence was edited. See lines 40-41
In order to work lines 66- SSL should be in brackets and the full name before it
R= Oberservation was taken into account. Sentence has been corrected. See line 67.
2.2. Starch extrusion - no such term "Regular maize" - native?
R= Oberservation was taken into account. Sentence has been corrected. See line 82.
line 75 "...were kindly provided by.." - there is a separate place in the publication for acknowledgments. There should only be statements here.
R= Oberservation was taken into account. Sentence has been corrected. See lines 77-79.
2.4. Starch and amylose contents - description too short, please extend the methodology
R= Oberservation was taken into account. Sentence has been corrected. See lines 115-125.
line 159- "against a distilled water blank" worth adding, whose T%=100..in brackets
R= Oberservation was taken into account. Sentence has been corrected. See line 176.
Very well described methodology regarding functional properties (emulsifiability and foam capacity and stability)
R= Thanks ¡
line 417"starch content" I don't understand, it's not a conclusion??????Please explain
In my opinion, the conclusions are too short and prove the purpose of the research undertaken.
R= Oberservation was taken into account. We reformulate the conclusions and include one related to the digestion profiles. See lines 430-439.
Reviewer 3 Report
The manuscript “Physicochemical and in vitro starch residual digestion structures of extruded maize and sorghum starches added with sodium stearoyl lactylate” is an interesting study with important results. Also, a statistician should revise the manuscript as the significant differences don’t seem correct. Otherwise, more discussions could be included in the results and discussions section.
In the introduction, there is too much information without references. Please revise and add proper references.
lines 53-56 – Please specify the other studies mentioned in this section.
line 65 – the term SSL should be specified at first use, as the abstract and the manuscript are considered separate texts
79 – please specify the type of extruder
lines 80 – 82 – this sentence seems unfinished, or correct where to were
lines 86, 88, etc. – please insert a space between the number and degree sign – revise the whole manuscript
section 2.4. – please at least indicate a reference to the calculation method
line 167 – gr – please correct and use the standard units of measurement
line 168 – the authors used 10000rpm, and at line 172, they used 1300*g – please use one type of rotation rate, and also at line 85 the authors used RPM with superscript, please revise the whole sections
line 202 – 203 – Please specify the producer and country of the production
table 3 – the letters should be put in superscript, and also please revise the statistics, 89.88±0.03a? are you sure of the correctness of the significant differences? Please revise. The letters don’t seem to be put correctly
Table 4 – there are some lines where no small letters are specified, and also, once again, the statistics and significant differences don’t seem correct. Please revise every table! and all the statistical analyses!
authors' contribution – please see the author's guideline and correct this section
However, the manuscript should be revised by a native English speaker for grammatical and orthographical errors.
Author Response
The manuscript “Physicochemical and in vitro starch residual digestion structures of extruded maize and sorghum starches added with sodium stearoyl lactylate” is an interesting study with important results. Also, a statistician should revise the manuscript as the significant differences don’t seem correct. Otherwise, more discussions could be included in the results and discussions section.
R= Discussion was updated in the revised manuscript. A revision on the statistical data was performed.
In the introduction, there is too much information without references. Please revise and add proper references.
R= Introduction was revised and proper references were updated.
lines 53-56 – Please specify the other studies mentioned in this section.
R= Sentence was edited in the revised version and the introduction was updated.
line 65 – the term SSL should be specified at first use, as the abstract and the manuscript are considered separate texts
R=Corrections were made in the revised version.
79 – please specify the type of extruder
R= Regular corn starch was extruded with a twin-screw corotating extruder (BCTM-30, Bühler, Switzerland) with a 600 mm length and L/D = 20 at a volumetric feed rate of 4 kg/h (calibrated with sugar; 8.50 kg of starch/h with a wet basis moisture content of 11.30%). Die diameter was 5 mm, and the screw configuration was selected specifically to create high levels of shear. The first section contained only conveying elements, with the next containing both conveying and kneading elements. Finally, the high-shear section contained conveying, reverse conveying, and kneading elements. Temperature was controlled at the final stage of the extruding chamber by using a TT- 137N water heater (Tool-temp, Sulgen, Switzerland). See lines 82-90.
lines 80 – 82 – this sentence seems unfinished, or correct where to were
R= We appreciate the revision. Sentence was edited in the revised version.
lines 86, 88, etc. – please insert a space between the number and degree sign – revise the whole manuscript
R= A complete revision was made thorough the manuscript.
section 2.4. – please at least indicate a reference to the calculation method
R= Reference was included, and the method briefly described in the revised version.
line 167 – gr – please correct and use the standard units of measurement
R= Units were thoroughly revised and corrected.
line 168 – the authors used 10,000rpm, and at line 172, they used 1300*g – please use one type of rotation rate, and also at line 85 the authors used RPM with superscript, please revise the whole sections
R= The section was edited to improve clarity. The 10,000 rpm was for homogenization using an Ultra Turrax device, and the 1,300 g was for centrifugation. The sentence was edited to clarify.
line 202 – 203 – Please specify the producer and country of the production
R= Corrections were made. See line 199.
table 3 – the letters should be put in superscript, and also please revise the statistics, 89.88±0.03a? are you sure of the correctness of the significant differences? Please revise. The letters don’t seem to be put correctly
R= Thanks for the observation. Significance letters were corrected in all tables and the statistical analysis revised.
Table 4 – there are some lines where no small letters are specified, and also, once again, the statistics and significant differences don’t seem correct. Please revise every table! and all the statistical analyses!
R= Letters were corrected and the statistical analysis revised.
authors' contribution – please see the author's guideline and correct this section
R= Guideline was followed in this section.
Round 2
Reviewer 1 Report
Dear authors,
thank you for revision.
Best regards
Reviewer 3 Report
The authors revised the manuscript quite well, however the letters of significant differences should be put with superscript.